# Improving Classification Accuracy of Hand Gesture Recognition Based on 60 GHz FMCW Radar with Deep Learning Domain Adaptation

**Hyo Ryun Lee [1] , Jihun Park [2] and Young-Joo Suh [2],\***

[1]   Department of Computer Science and Engineering (CSE), Pohang University of Science and
    Technology (POSTECH), Pohang 37673, Korea; wowlhr@postech.ac.kr
[2]   Graduate School of Artificial Intelligence (GSAI), Pohang University of Science and Technology (POSTECH),
    Pohang 37673, Korea; lorenzopark@postech.ac.kr
\*   Correspondence: yjsuh@postech.ac.kr; Tel.: +82-054-279-2243

**Abstract:** With the recent development of small radars with high resolution, various human–computer interaction (HCI) applications using them have been developed. In particular, a method of applying a user's hand gesture recognition using a short-range radar to an electronic device is being actively studied. In general, the time delay and Doppler shift characteristics that occur when a transmitted signal that is reflected off an object returns are classified through deep learning to recognize the motion. However, the main obstacle in the commercialization of radar-based hand gesture recognition is that even for the same type of hand gesture, recognition accuracy is degraded due to a slight difference in movement for each individual user. To solve this problem, in this paper, the domain adaptation is applied to hand gesture recognition to minimize the differences among users' gesture information in the learning and the use stage. To verify the effectiveness of domain adaptation, a domain discriminator that cheats the classifier was applied to a deep learning network with a convolutional neural network (CNN) structure. Seven different hand gesture data were collected for 10 participants and used for learning, and the hand gestures of 10 users that were not included in the training data were input to confirm the recognition accuracy of an average of 98.8%.

**Keywords:** 60 GHz FMCW radar; deep learning; domain adaptation; hand gesture recognition; human activity recognition (HAR)

---

## 1. Introduction

In recent years, with the remarkable development of smart devices, human activity recognition (HAR) technology is being actively applied in various fields such as entertainment, healthcare, security, public safety, industry, and autonomous vehicles [1]. Accordingly, user actions and gestures recognition systems using signals from wireless communication technologies such as Wi-Fi, ultra-wide band (UWB), and Bluetooth, as well as vision cameras, are being studied in various ways. Among them, the hand gesture recognition system has been in the spotlight as an input device in HCI. It is being applied to computer games, virtual reality (VR) contents, and non-contact appliance controllers. The traditional hand gesture system using a vision camera [2–4] shows high recognition accuracy, but there is the drawback of showing a sharp decline in accuracy in a low-light environment or when an obstacle obstructs the camera. Another traditional method, a motion recognition system in which a wearable device is attached to the body [5], makes the user feel uncomfortable.

In order to overcome the disadvantages of vision cameras and wearable devices, activity and gesture recognition systems using wireless communication signals such as Wi-Fi, Bluetooth, UWB,

and ZigBee have been attempted in various ways in recent years [6–13]. These wireless signals can capture motions or gestures even when there are obstacles around the device due to characteristics such as diffraction and penetration. However, a motion recognition system based on a wireless signal is greatly affected by the surrounding environment such as signal attenuation and multipath fading problems [14]. Furthermore, it does not have sufficient distance resolution to distinguish motion or gesture due to a relatively narrow bandwidth. To solve this problem, a Wi-Fi device-based radar is proposed to secure the resolution to distinguish hand movement in [15]. However, the maximum bandwidth by the Wi-Fi standard and additional antenna requirements are still disadvantages.

Recently, problems with various conventional systems have been solved through deep learning. Deep learning is one of the subsets of artificial intelligence and it extracts vast amounts of information about the problem. Then, it plays a big role in finding a solution by grasping a very complex correlation between the extracted information. Performance improvement through the application of deep learning has successfully been applied to systems in various fields such as medical devices, indoor positioning, and VR [16–19]. The DeepHandsVR [16] proposed hand interface to deep learning was applied. A deep learning model based on CNN was designed. It learned the input of the VR controller and inferred the gesture image quickly and accurately. In [17], a CNN-based deep learning model was also used, and an imaged range of UWB signals was trained. Through this, improved localization accuracy was shown. The PERI platform [18] analyzed users' purchase intention, interest, and impact on purchase in various ways using a deep learning model in an E-commerce platform. In [19], a scheme for arrhythmia prediction through deep learning analysis of an electrocardiogram (ECG) heartbeat was proposed. It used two types of auto-encoders in the training phase, and a fine-tuned deep neural network (DNN) was used as a classifier. In this way, deep learning is applied to various applications to solve problems that humans cannot easily analyze.

In addition, deep learning is being applied in various ways to small frequency modulated continuous wave (FMCW) radar modules that are being actively developed in recent years. Accordingly, motion and gesture recognition systems based on the high resolution of distance and velocity are being studied using a very wide bandwidth of the FMCW radar [20–25]. Due to the characteristics of the radar signal, it is free from the low-light problem of the vision camera, and the inconvenience experienced by the user, which is a problem of wearable devices, can be solved. In addition, unlike a system based on a wireless communication signal, very high bandwidth is used, and then, it has sufficient distance and velocity resolution for distinguishing motion. Therefore, a small FMCW radar is suitable for recognizing motions and gestures. In [20], user motion was classified by a deep learning model based on a random forest algorithm using Doppler images of a 60 GHz radar. In [21], a radar operating at 5.8 GHz and a Doppler spectrogram were used. This system classified hand gestures through a deep learning network composed of several CNN layers. It is similar to the previous system, but higher gesture recognition accuracy was shown in [22] by using a 24 GHz radar. Latern [23] proposed a deep learning model that combines 3D-CNN layers and Long Short Term Memory (LSTM) networks for continuous hand gesture recognition. In [24] and [25], the temporal features of gestures based on 24 GHz radar signals were extracted. Using these as input data of the LSTM-based deep learning model, the gesture recognition result was inferred. Additionally, a smartphone, called Google Pixel 4, with a built-in Soli radar module has been released, enabling device control, called MotionSense, through hand gestures at close range.

For the above reasons, this paper proposed a hand gesture recognition system using an FMCW radar, called Hatvan, operating in the 60 GHz band. Existing studies using an FMCW radar in the 24 GHz band [22–24] have relatively long object detection distances, but it is difficult to recognize precise motion. In contrast, the 60 GHz FMCW radar used in this paper has sophisticated detection capability for very small hand gesture changes by utilizing a higher bandwidth. In addition, the proposed gesture recognition system used deep learning, a general framework for motion recognition method these days. The Range–Doppler Matrix (RDM), a feature from the hand gesture movements, is learned through a deep neural network with a CNN structure. When a hand gesture is provided as an input,

RDM is extracted and the recognition result is inferred. However, such a general deep learning-based gesture recognition system has a common problem. Even though the same types of hand gesture are taken, the difference in movements that occur depending on the user affects the inference result. In other words, due to the difference between the gesture information used in the learning stage and the gesture information input from the user in actual use, the accuracy of classifying results by the deep neural network is degraded.

To solve this problem, this paper introduced a domain adaptation algorithm to the hand gesture recognition system. Domain adaptation adapts information of an existing domain when data are input from a new domain (target domain) different from the existing domain (source domain) in which the model was successfully operated [26–34]. In detail, deep domain confusion (DDC) [26] and Deep CORAL [27] defined metrics representing distance between different two domains. Based on the distance metric, DDC minimized the discrepancy on the feature space between the two domains by using an additional fully connected adaptation layer. Similarly, Deep CORAL minimized the Frobenius norm between the covariance matrices of the two domains. The most common method in domain adaptation is to apply adversarial training. It uses a domain classifier that can distinguish between two different domains. The gradient derived by the domain classifiers is transferred in reverse to the feature extractor. As a result, the feature extractor is unable to distinguish between the two domains. Accordingly, the entire deep learning network can well infer the results for any data. In [28–32], they differ only in the method of deriving the loss of the domain classifier, but follow the general method of adversarial training. Furthermore, studies to improve domain adaptation performance were also conducted. Decision-boundary Iterative Refinement Training with a Teacher (DIRT-T) [33] used entropy minimization to prevent the decision boundary of the classifier from violating the clustering assumption. In [34], domain adaptation performance is improved by applying a self-supervision that adds an auxiliary task of creating its own labels directly from the data. Representative methods for domain adaptation will be covered in detail in Section 2.2.

In the hand gesture recognition system, the source domain corresponds to the gesture data collected in the learning step, and the target domain corresponds to the gesture data from users whose data are not used as training data to be classified through the learning model. Even though each person makes the same gesture, there is a difference in movement such as swipe angle and distance, rotation radius, and push intensity, so the distribution of data collected for each user varies, and then, the extracted characteristics change accordingly. Therefore, the difference in fine motion causes a difference in the feature space applied to the deep neural network model, and consequently, the inference performance is degraded. Moreover, since it is impossible to learn all the differences in motions that are different for each person, unsupervised domain adaptation is used to accurately classify unlearned user gesture data.

The remainder of this paper is organized as follows. Section 2 introduces preliminaries related to 60 GHz FMCW radar signal processing and domain adaptation to facilitate understanding. In Section 3, the hand gesture recognition system with improvement of classification accuracy through domain adaptation is proposed. The gesture recognition experiment setup including 60 GHz FMCW radar and data used in experiment are introduced in Section 4. In Section 5, we analyze the experiments and evaluate the results. Finally, we conclude this paper in Section 6.

## 2. Preliminaries to Radar-Based Gesture Recognition and Domain Adaptation

*2.1. 60 GHz FMCW Radar*

2.1.1. Radar System Overview

The FMCW radar used for hand gesture recognition was a Hatvan module manufactured by Infineon, which is similar to the Google Soli [35,36] module. The radar functions are mostly similar, but Soli has four receiving antennas, whereas Hatvan has three receiving antennas with a transmitting antenna. This radar module is operated in the 60 GHz unlicensed band and has a resolution in

centimeters through the use of millimeter waves (mmWave), making it suitable for hand gesture recognition. Since it is operating in the V-band and can transmit a chirp signal of up to 6 GHz (57.5–63.5 GHz) bandwidth, it can secure a range resolution of 2.5 cm and a Doppler resolution of about 122 cm/s can be obtained according to Equations (1) and (2), respectively.

$$\Delta r = \frac{c}{2B} = 2.5 \text{ cm,} \tag{1}$$

$$\Delta v = \frac{c}{2 f_c} \cdot \frac{1}{lT} \cong 122 \text{ cm/s,} \tag{2}$$

where $c$ is the speed of light, approximately $3 \times 10^8$ m/s, and $f_c$ is set to 60 GHz, which is the center frequency between 57.5 and 63.5 GHz. Accordingly, $B$ is calculated as 6 GHz, which is the bandwidth of a chirp signal. $T$ is the sweep timing of the FMCW radar, $l$ is the number of repeatedly transmitted chirp signals in a shape group, and then these are set to 128 μs and 16, respectively.

The FMCW radar sends a periodic chirp signal through a transmitting antenna, and receives a signal reflected from an object through multiple receiving antennas. The transmitted chirp signal is frequency modulated by a periodic sawtooth wave function and a time delay $\tau$ and Doppler shift $f_d$ occur with the signal received by reflection, as shown in Figure 1.

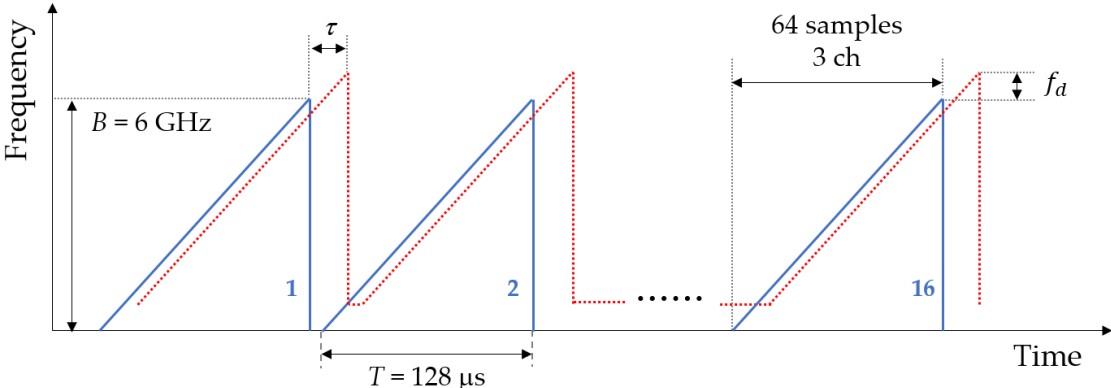

**Figure 1.** Frequency modulated continuous wave (FMCW) waveform in the frequency domain for a shape group.

In Figure 1, the blue solid line and the red dotted line represent the transmitted signal and reflected signal, respectively. Sixty-four samples are acquired from a chirp signal, and 16 chirps from one shape group. The four shape groups are repeatedly transmitted in one frame to obtain information reflected from the object.

The time delay $\tau$ is caused by the distance between the radar and the object reflecting the signal, and the Doppler shift $f_d$ is caused by the movement of the object moving away or closer to the radar [37]. Using these characteristics, the radar system can recognize which gesture the user took by analyzing the time delay $\tau$ and Doppler shift $f_d$ changes that occur in accordance with the hand movement.

### 2.1.2. Radar Signal Processing for Gesture Recognition

In general, when using a wireless transceiver device, if raw signals are used as they are, it is very difficult to extract the desired feature due to the surrounding environment or white noise. Particularly, in the case of hand gesture recognition using an FMCW radar, the wave forms for different behaviors based on raw signals expressed on the time–amplitude plane are not clearly distinguished. To solve this problem, general radar-based motion recognition applications perform 2D fast Fourier transform (FFT) to extract a distance–speed map of an object. Especially, if a zero-padding-based 2D FFT is applied, more precise distance and Doppler shift accuracy can be obtained because more frequency

bins are generated after transformation [38]. Accordingly, the resolution of the range and Doppler bins after applying a 2D FFT may be defined as follows:

$$\Delta r_f = \frac{c}{2B} \cdot \frac{f_s}{N/T} = 0.31 \text{ cm,} \tag{3}$$

$$\Delta v_f = \frac{c}{2f_c} \cdot \frac{1}{LT} = 7.63 \text{ cm/s,} \tag{4}$$

where $f_s$ is set to 500 kHz as the sampling frequency and the size of 2D FFT ($N \times L$) is set to $512 \times 256$ so that the radar system has an appropriate resolution value to recognize the hand gesture.

By applying 2D FFT to the reflected signal from the FMCW radar mentioned in the previous subsection, a range–Doppler can be obtained [39], as shown in the following equation.

$$S(p, q, t) = \sum_{l=0}^{L}\left(\sum_{i=0}^{n} s(n, l, t)e^{-j2\pi pn/N}\right)e^{-2\pi ql/L}, \tag{5}$$

$$RD(r, v, t) = \left|S\left(\frac{r}{\Delta r_f}, \frac{v}{\Delta v_f}, t\right)\right|, \tag{6}$$

where $s(n, l, t)$ is the beat signal [40], which is first transformed, corresponding to the transmitted chirp signal and this signal is transformed to the frequency domain to obtain Doppler–FFT expressed in $S(p, q, t)$ [41]. In other words, $S(p, q, t)$ is an output matrix at frame $t$ in the frequency domain after 2D FFT and every axis represents the range and Doppler between the radar and an object.

The Range–Doppler Matrix (RDM) can be obtained from $S(p, q, t)$ and expressed as Equation (6). As a result, RDM that includes the range and radial velocity, which is derived from Doppler, of a target object and undesired clutter or background noise can be expressed as follows:

$$RDM = \begin{bmatrix} \widetilde{S}(1, 1) & \widetilde{S}(1, 2) & \dots & \widetilde{S}(N_C, 1) \\ \widetilde{S}(1, 2) & \widetilde{S}(2, 2) & \dots & \widetilde{S}(N_C, 2) \\ \vdots & \vdots & \ddots & \vdots \\ \widetilde{S}(1, N_S) & \widetilde{S}(2, N_S) & \dots & \widetilde{S}(N_C, N_S) \end{bmatrix}, \tag{7}$$

where $N_C$ is the number of Doppler–FFT points and $N_S$ is the number of range–FFT points. Based on Equation (5), $q$th Doppler–FFT output $\check{S}(q, k)$ using the index of range bins in the range domain $k$ is defined as follows:

$$\widetilde{S}(q, k) = \sum_{l=1}^{N_C} S(l, k)e^{-2\pi(l-1)(q-1)/N_C}, \tag{8}$$

As mentioned above, the RDM expressed in Equation (7) includes information on signals reflected by other uninterested objects as well as hand gesture. Generally, a hand gesture recognition environment using radar assumes an environment in which the radar and the hand are at a relatively close distance. This means that the RDM change according to the hand movement is the largest, and the RDM change due to the uninterested object is very small. As a result, if a background subtraction method can be applied to a stationary object, only clutter caused by hand movement can be extracted. For this reason, an adaptive background model based on the most widely known Gaussian mixture model was used [42]. After comparing the background model with the current frame of RDM, only clutter caused by hand movement was extracted by performing background subtraction, which removes clutter caused by static objects or the environment.

In addition to RDM extraction of the movement of interest through background subtraction, an important consideration is to detect and classify only when there is actually a gesture. If this is not taken into account, it tries to classify the RDM by a fine clutter close to it from the radar, which acts as a

major cause of degrading the overall performance of the hand gesture recognition system. A constant false alarm rate (CFAR) algorithm is commonly used in target detection and radar signal processing to solve this problem [43–45]. In this paper, the CFAR algorithm used in [46] is applied to simplify signal processing and reduce signal processing overload. This CFAR algorithm calculates the moving average using an exponentially weighted moving average (EWMA) and determines that there is an actual hand gesture only when the raw signal received by the radar module exceeds a certain threshold.

$$x_t = \sum_i \|RD^i(r, \, v, \, t-1)\| , \tag{9}$$

$$M_t = (1 - \alpha)M_{t-1} + \alpha x_t, \tag{10}$$

In Equation (9), $x_t$ is the sum of all pixel values on the RDM created by three channels through receiving antennas and $RD^i$ is the matrix of RDM for $i$th channel based on Equation (6). Using $x_t$ with constant smoothing factor $\alpha$, moving average is derived. The mathematical expression of the condition exceeding a certain threshold by comparing the moving average derived from Equation (10) and the raw signal is as follows:

$$|x_t - M_t| > \theta \cdot \left( M_t + M_{offset} \right), \tag{11}$$

where $\theta$ is a detection threshold, and $M_{offset}$ is an offset parameter, and the radar system's gesture detection sensitivity is adjusted according to the settings of the two coefficients. As a result, only the RDM extracted from the raw signal of the condition satisfying Equation (11) is used for gesture classification and the raw signal corresponding to other conditions is ignored.

To summarize the overall operation of the radar module, it is shown in Figure 2. When a user moves their hand at a close distance from the radar, the chirp signal from the transmitting antenna is reflected by the hand. Then, the reflected raw signal is collected by the receiving antennas. Two-dimensional FFT is applied to extract features of distance and radial velocity. Unnecessary information from non-interested objects other than the hand is removed by background subtraction. The CFAR algorithm finds the actual hand gesture, and RDMs to classify the type of gesture are extracted. Figure 3 shows examples of the RDM snapshot related to each gesture class.

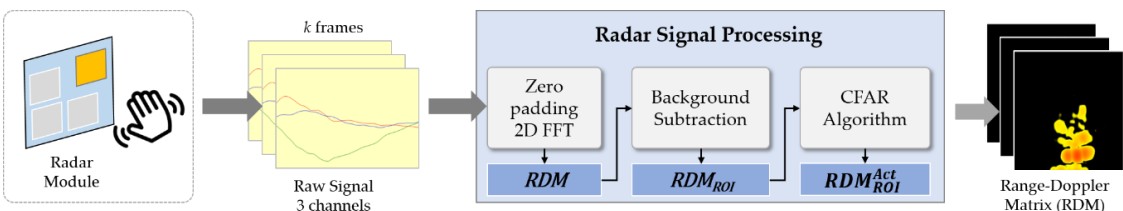

**Figure 2.** Overview of the Range–Doppler Matrix (RDM) extraction process for hand gesture.

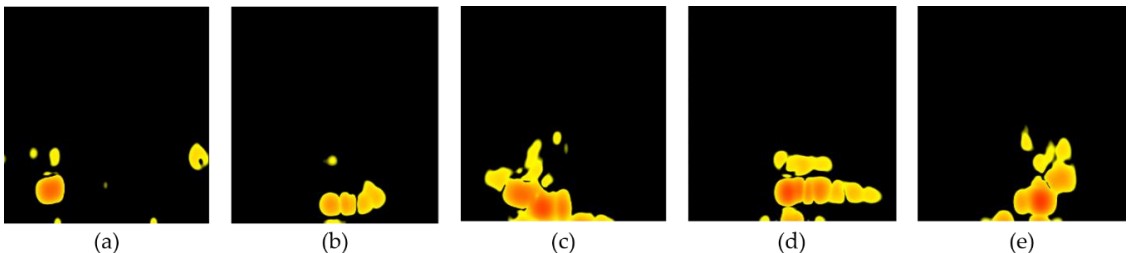

**Figure 3.** RDM visualization example according to actual hand gestures in a 10–15 cm range. Horizontal axis means target radial velocity, vertical axis means range, and color represents decibels relative to full scale (dBFS): (**a**) Swipe; (**b**) Grab; (**c**) Push; (**d**) Clockwise; (**e**) Double-push.

## 2.2. Domain Adaptation

Domain adaptation is used to effectively infer in the target domain where the label is insufficient or does not exist [29]. When the distributions of training data (source domain) and test data (target domain) are different, domain adaptation enables efficient inference by adjusting the discrepancy of the new test data domain (target knowledge) to be similar to the training data domain distribution (source knowledge). In other words, domain adaptation is a method of improving the inference performance with relatively small labeled or unlabeled data collected in the target domain data by using a source domain that has a rich label and is related to the target domain.

To express unsupervised domain adaption mathematically, every classifier $\eta$ is defined as a function that maps from input space X to label space Y as follows:

$$\eta: X \rightarrow Y, \tag{12}$$

The source domain $D_S \subset X \times Y$ is defined as a set of samples $x_i^S \in X$ sampled from the random variable $X^S$ and labels $y_i^S \in Y$ as follows:

$$D_S = \left\{ \left( x_i^S, \, y_i^S \right) \right\}_{i=1}^N, \tag{13}$$

Similarly, the target domain $D_T \subset X \times Y$ is defined as a set of unlabeled target samples from random variable $X^T$; then, the input of the target domain is expressed as

$$D_T^X = \left\{ x_i^T \right\}_{i=1}^M, \tag{14}$$

Based on the above definition, as shown in Equation (15), the domain adaptation task is to find a proper classifier $\eta$ that can minimize the risk function $R_{D_T}(\eta)$ for the target domain. This is made possible by lowering the risk by training so that the target domain and the source domain cannot be distinguished as much as possible. In other words, this method makes it possible to create a classifier that works properly in test data sampled from the target domain.

$$R_{D_T}(\eta) = Pr_{(x,y) \sim D_T}(\eta(x) \neq y), \tag{15}$$

As a result, for efficient classification using domain adaptation, it is necessary to create and train a model that has high classification performance in the source domain and cannot distinguish between the source domain and the target domain.

In this way, methods of improving the data classification performance of the target domain by applying domain adaptation in various fields have been studied. One way of domain adaptation is training the models to minimize Euclidean distances or other metrics indicating the quantitative difference between the source and target domain samples in feature space. Deep CORAL [27] is one of such methods minimizing CORAL loss, a differentiable loss based on the second-order statistics. Its integration is performed by adding a few adaptation layers at the end of the networks as shown in Figure 4. The framework consists of two networks, one for the source domain and the other for the target domain, and the networks are jointly optimized by the CORAL loss along with the typical classification loss in an end-to-end manner.

Domain adaptation can be achieved by an adversarial training approach. Representatively, the domain-adversarial neural network (DANN) [28] suggests domain adaptation by introducing a domain discriminator and gradient reversal layer to the existing neural network structure including a feature extractor and classifier as shown in Figure 5. The domain discriminator is a domain classifier cheated by a gradient reversal layer that makes the source domain and the target domain indistinguishable. The gradient of confusion alignment loss is propagated backwards through the

gradient reversal layer and the feature map obtains negative feedback for the domain distinguishable feature output, so that the source domain and the target domain cannot be distinguished.

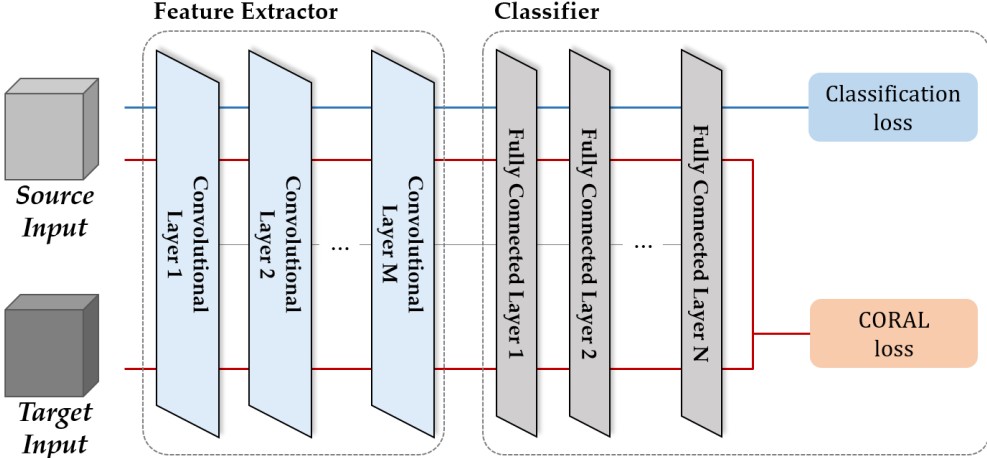

**Figure 4.** The architecture of the Deep CORAL domain adaptation.

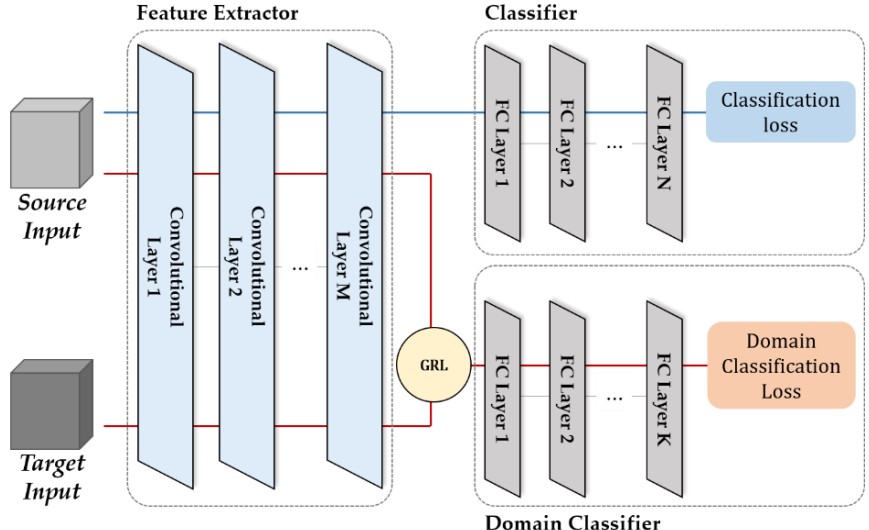

**Figure 5.** The architecture of domain-adversarial neural network (DANN) domain adaptation.

Decision-boundary Iterative Refinement Training with a Teacher (DIRT-T) [33] is introduced along with virtual adversarial domain adaptation (VADA) to resolve two critical issues and drastically improves target performance. DANN may fail to work if (1) the feature extractor has too high capacity or (2) the domain shift is too large to work well on both domains. Training with source domain data, VADA adopts the cluster assumption to prevent decision boundaries crossing a dense data region via entropy minimization and integrates with Lipschitz constraints for reliable prediction results. DIRT-T, as shown in Figure 6, iteratively refines the VADA-initialized decision boundary, updating one step from the previous step as a teacher, towards achieving the minimum value of conditional entropy.

Recently, self-supervision provides a new direction for domain adaptation [34]. It proves that the self-supervision method with auxiliary tasks helps the target adaptation performance. It argues that well-designed auxiliary self-supervision tasks can help to capture the structural information of the images. It manually constructs datasets for auxiliary tasks, such as rotation prediction, flip prediction, and patch location prediction, from the given source and target image dataset. In Figure 7, the proposed architecture has a feature extractor and it is shared by multiple heads, which are the supervised main task classifier and self-supervised auxiliary task classifiers. The multi-task training is applied for training them all at once.

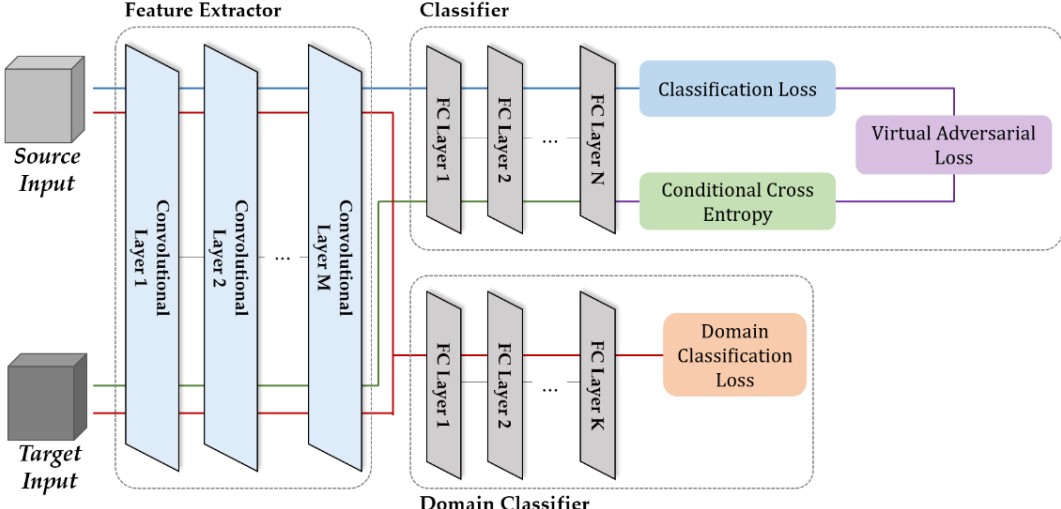

**Figure 6.** The architecture of Decision-boundary Iterative Refinement Training with a Teacher (DIRT-T) domain adaptation.

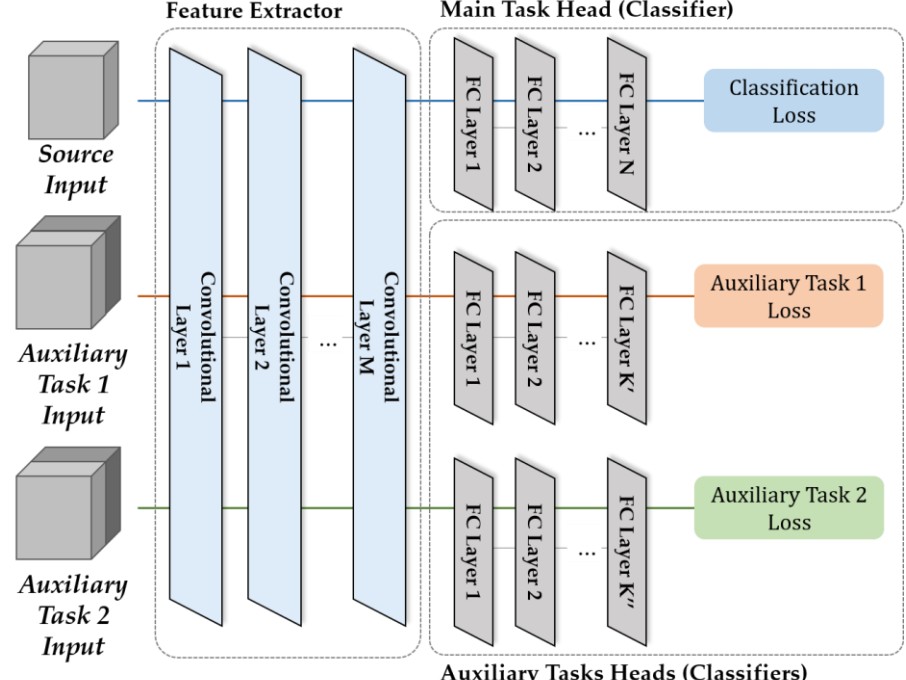

**Figure 7.** The architecture of self-supervision-based domain adaptation.

## 3. Hand Gesture Recognition System

### 3.1. System Overview

This paper proposes a network structure, as shown in Figure 8, to improve the accuracy of hand gesture recognition based on 60 GHz FMCW radar. A set of the Range–Doppler matrix derived by signal processing in the preliminaries section (Section 2.1.2) is input to the machine learning network. Then, spatial–temporal features related to distance and radial velocity are extracted by a 3D-CNN-based feature extractor. The extracted feature is input to a hand gesture recognizer to infer the gesture result or input to a domain discriminator to learn the data on the target domain in an adversarial way.

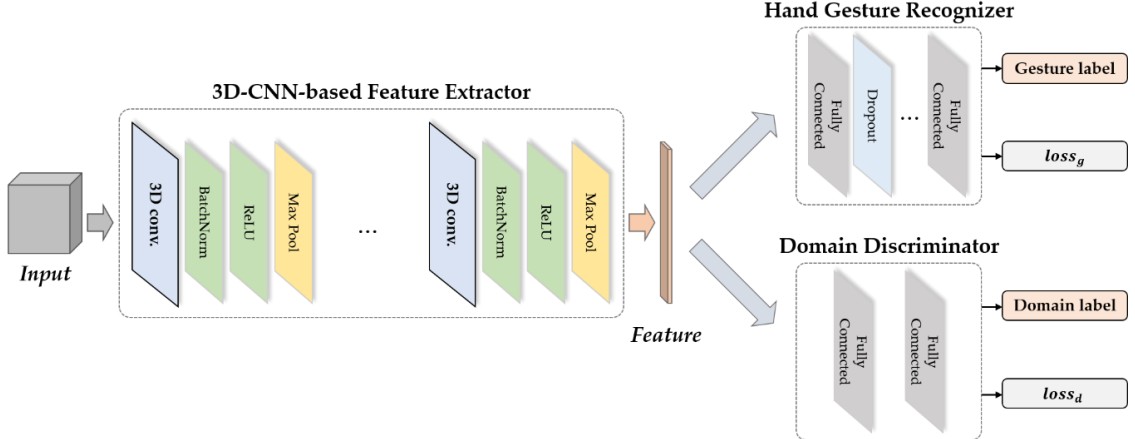

**Figure 8.** The overview of the proposed hand gesture recognition system.

### 3.2. Input Dataset

In the FMCW radar-based hand gesture recognition system, gestures can be classified using various types of input. In the case of Google Soli [32], range profiles and Doppler profiles are generated from the Range–Doppler derived in Equation (6), and a motion profile is created by concatenating them and is inputted to the CNN as follows:

$$RP_t^i(r,\ t) = \sum_v RD^i(r,\ v,\ t), \tag{16}$$

$$DP_t^i(v,\ t) = \sum_r RD^i(r,\ v,\ t), \tag{17}$$

$$MP_t = \left(RP_t^1, RP_t^2, RP_t^3, DP_t^1, DP_t^2, DP_t^3\right), \tag{18}$$

where $i = 1, 2, 3$, which indicates the $i$th receiver antenna. $RD^i(r,\ v,\ t)$ is the Range–Doppler for $i$th receiver at frame $t$. This solves the problem of requiring a lot of computation costs when using RDM sequences directly and thus, making real-time gesture recognition applications. However, when a motion profile with a relatively low computational load is used as an input, there is a disadvantage that the gesture recognition accuracy is slightly degraded due to the increase in the number of types of hand gesture or the ambiguous direction in operations such as rotation and drawing of a figure.

On the other hand, if the Doppler spectrogram as a micro-Doppler signature [21] is used as an input for machine learning, the classification accuracy is very low unless the hand gesture is captured at the exact point in time. That is, when the gesture recognition system captures the spectrogram from the middle part of the gesture and the motion information is fragmented, it is difficult for the model to properly infer the recognition result.

In order to compensate for the problems of the two types of inputs mentioned above, in this paper, RDMs derived from raw signals received from each receiving antenna are concatenated to generate one integrated RDM. By using the CFAR algorithm, the integrated RDMs during the time in which the gesture occurs are input into the learning model, as shown in Equation (19).

$$RDM_{INT} = \left(RDM_t^1, RDM_t^2, RDM_t^3\right), \tag{19}$$

The process of generating the input dataset explained above can be described as shown in Figure 9. During the frame time when the user makes a gesture close to the FMCW radar, three receiving antennas derive RDM frames from raw signals. The RDMs from each frame are concatenated into one integrated RDM, and an input dataset is formed by collecting the integrated RDMs for the time taken by the gesture. The generated input data are fed to a CNN-based feature extractor.

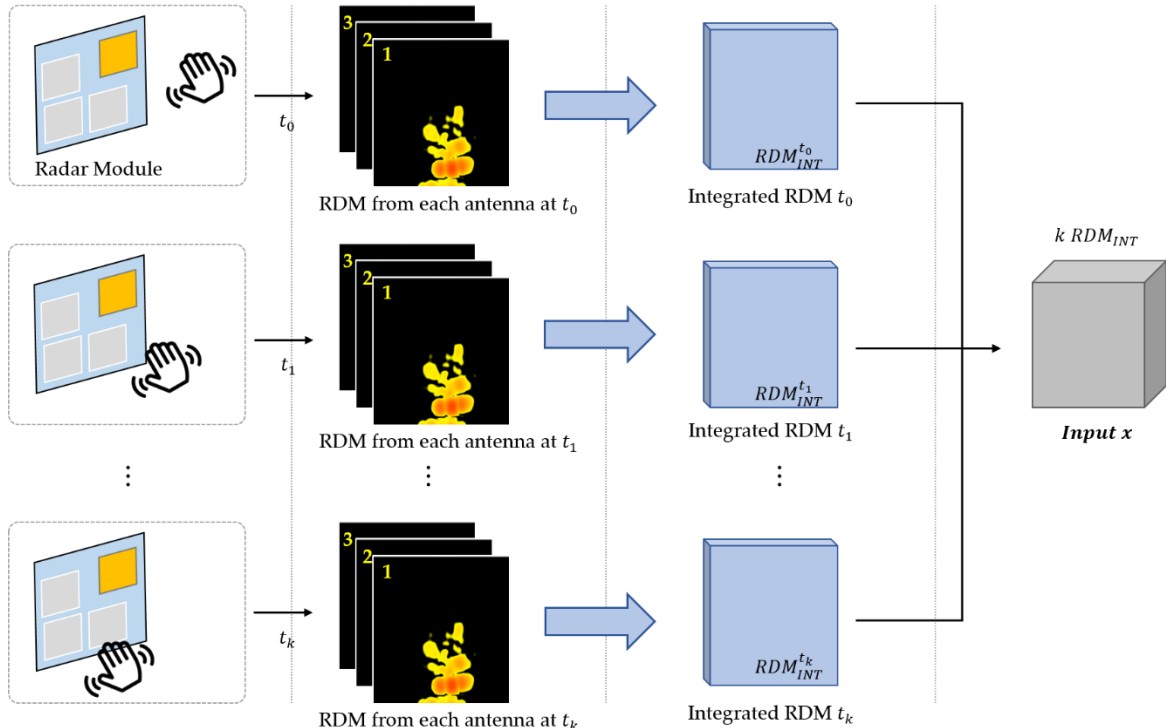

**Figure 9.** The overview of input dataset collection.

### 3.3. Feature Extractor

The network architecture for feature extraction is shown in Figure 10 in the hand gesture recognition system. If the spatial movement as well as the distance and velocity of the hand movement are considered, the accuracy of gesture recognition can be improved, and for this, a spatial–temporal feature analysis of a gesture is required. Accordingly, the feature extractor in this paper is based on a 3D-CNN structure for short spatial–temporal modeling. However, the commonly used 3D-CNN structure is not suitable for applications requiring real-time performance such as motion recognition due to the disadvantages of many parameters and excessive computation load. In general, 3D convolution is the main cause of the most demanding computational load and parameter increase in the 3D-CNN structure. Therefore, in this paper, a convolution block with an improved 3D convolution mechanism is proposed and used as shown in Figure 11.

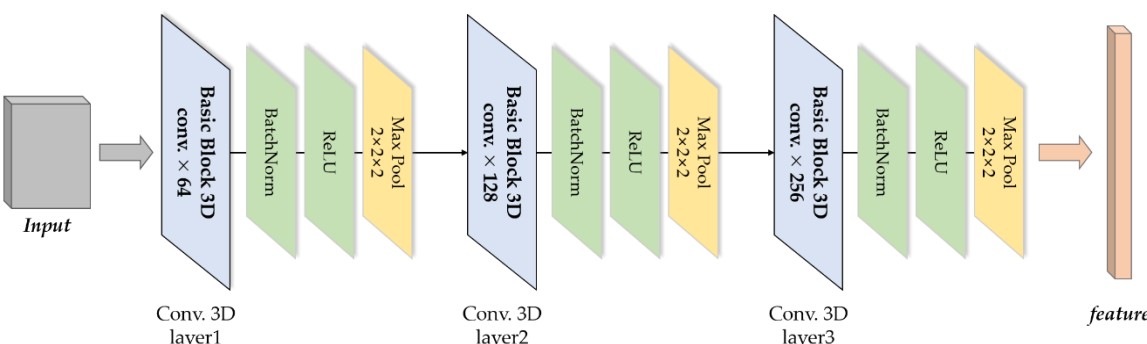

**Figure 10.** Architecture of a feature extractor based on 3D-CNN.

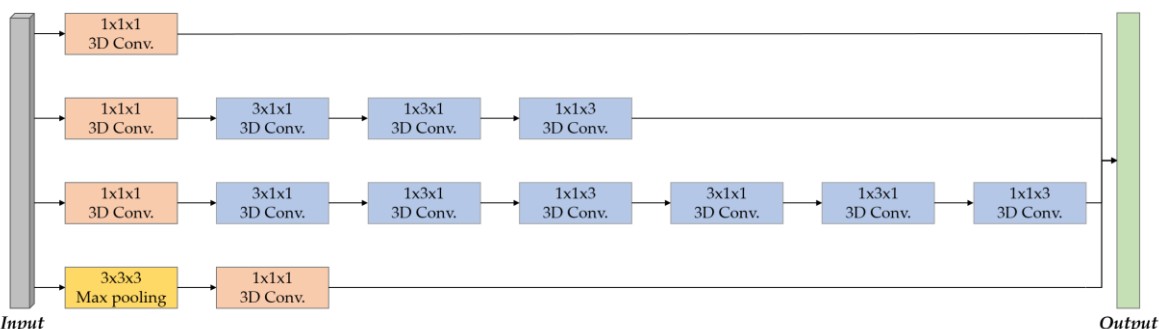

**Figure 11.** The architecture of a basic block for 3D-CNN.

In order to improve the 3D convolution operation speed and reduce the number of parameters, the Inception structure used by GoogLeNet [47], which won the 2014 ImageNet classification competition, was applied to a convolution filter to decompose it. The Inception structure is characterized by the parallel use of various filters and convolution to reduce the number of channels, as shown in Figure 12a. If the Inception structure is applied to the 3D convolution filter commonly used in 3D-CNN, it can be decomposed, as shown in Figure 12b. Based on this, by extending the Inception structure to 3D and introducing it to the 3D convolution filter structure, it can be made into a basic block of 3D-CNN, as shown in Figure 11, thereby increasing the feature extraction efficiency by reducing the amount of computation required for the feature extractor network.

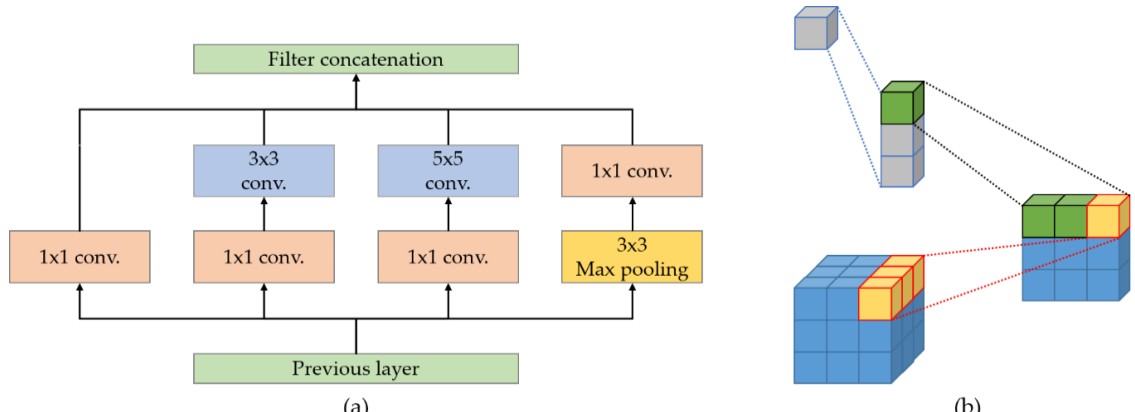

(a)                                                                    (b)

**Figure 12.** 3D convolution filter with Inception structure. (**a**) The architecture of GoogLeNet Inception; (**b**) Decomposition principle of convolution filter using Inception structure.

The 3D-CNN-based feature extractor network (Figure 10) using basic blocks with improved computation speed is used to learn the spatial and temporal features of consecutive short frames. Even if the structure of the existing 3D convolution filter is modified, the effect of applying the convolution kernel cube is the same because the output for the input of the basic block is the same. Therefore, according to the general 3D convolution operation [48], the value at the $(x, y, z)$ position in each feature cube can be formulated as follows:

$$Y(x, y, z) = X(x, y, z) \otimes H(x, y, z)$$
$$Y(x, y, z) = \sum_{i=0}^{K_1-1} \sum_{j=0}^{K_2-1} \sum_{k=0}^{K_3-1} X(x+i, y+j, z+k)H(i, j, k), \tag{20}$$

where $X(x, y, z)$ and $Y(x, y, z)$ stand for the adjacent previous frame at the $(x, y, z)$ position and each feature cube, respectively. $H(x, y, z)$ indicates a convolution kernel cube, and the basic block replaces this general 3D kernel. $K_1$, $K_2$, and $K_3$ denote the length, width, and height of the convolution kernel cube, respectively. Like the general network structure of 2D-CNN, pooling is applied to 3D-CNN,

and through this process, the same number of feature cubes with reduced spatial and temporal resolution are created. In addition, batch normalization [49] is applied between the convolution layer and the pooling layer as in [24] to accelerate the learning network, and an activation function follows with the batch normalization layer. The feature extractor employs Rectified Linear Units (ReLU) to introduce non-linearity. The feature obtained from the proposed 3D-CNN-based feature extractor is expressed as the following with the input data $x$ and parameters of feature extractor $\theta_d$:

$$G_f(x) = 3DCNN(x; \theta_d), \tag{21}$$

### 3.4. Gesture Recognizer

In the gesture recognizer, it trains a gesture classifier using the source domain features with associated labels. The gesture recognizer is connected to the feature extractor and consists of three fully connected layers and a dropout layer. Generally, the parameter increases excessively in the fully connected layer, causing an overfitting problem, which significantly reduces the generalization performance. Therefore, dropout [50] is applied to reduce generalization error. Since the last fully connected layer performs the hand gesture recognition function and corresponds to the learning of the source domain, the loss can be expressed in the form of cross-entropy as follows:

$$loss_g = -\frac{1}{n_s} \sum_{x_{s_i} \in D_S} \sum_{n=1}^{N} l_{s_in} log G_g\big(G_f\big(x_{s_i}\big)\big), \tag{22}$$

where $n_s$, $N$, and $l_{s_in}$ stand for the amount of data in the source domain, the number of gesture classes, and binary variable which indicates whether $s_i$th data belong to the $n$th class. $x_{s_i}$ is source domain data with the corresponding label and $D_S$ indicates source domain. $G_g$ denotes the gesture recognizer and $G_f$ denotes the feature extractor.

### 3.5. Domain Discriminator

The purpose of the domain discriminator is elimination of the discrepancy between the source domain and target domain, thus a domainindependent feature is obtained for classification. As explained in the preliminaries section (Section 2.2), in this paper, a domain discriminator learns in an adversarial manner such as DANN [24] among various methods of domain adaptation. Through adversarial learning, high recognition accuracy can be achieved even for unlabeled target domain data by using features that are independent of the domain to cheat the classifier. That is, feature $F_T$, obtained by inputting target domain data into the feature extractor, is extracted, and distinguished through feature $F_S$ for the source domain data. The output of the domain discriminator is the probability that the input target domain data belongs to the source domain. If the loss, which is based on output result, is back-propagated and trained in an adversarial manner, the distinction between the source domain and target domain becomes ambiguous; the data from the target domain can be properly classified.

Similar to the gesture recognizer, the domain discriminator consists of two fully connected layers and the last fully connected layer performs the domain classification function. In addition, the loss of the domain discriminator can be expressed in the form of cross-entropy as follows:

$$loss_d = -\frac{1}{n_s + n_t} \sum_{x_{s_i} \in D_S \cup D_T} \sum_{m=1}^{M} d_{im} log G_d\big(G_f\big(x_{s_i}\big)\big), \tag{23}$$

where $n_s$ and $n_t$ stand for the amount of data in the source domain and target domain, respectively. $M$ is the number of domains and $d_{im}$ is a binary variable that indicates whether $i$th data belong to the $m$th domain. $D_T$, $G_g$, and $G_f$ denote target domain, the gesture recognizer, and feature extractor,

accordingly. By combining Equations (22) and (23), overall loss of the network can be obtained as follows:

$$loss = loss_g - \alpha loss_d, \tag{24}$$

where $\alpha$ is the trade-off coefficient and the final goal of the proposed hand gesture recognition system is to minimize overall loss by minimize the $loss_g$ for maximum gesture recognition accuracy and maximize the $loss_d$ to obtain domain-independent features.

## 4. Experiments

### 4.1. Experimental Setup

For the verification of the proposed hand gesture recognition system, we use a Hatvan radar module manufactured by Infineon, which has a transmitting (Tx.) antenna and 3 receiving (Rx.) antennas, as shown in Figure 13a. It operates in 60 GHz unlicensed band and covers from 57 to 64 GHz. It has 3 ADC channels with 12 bits resolution and up to 3.3 MSps sampling rate to sample the RX-IF channels. Its baseband chain consists of a high pass filter, low noise voltage gain amplifier (VGA), and antialiasing filters. This FMCW radar is connected to a laptop via USB and delivers raw signal information received from 3 Rx, as shown in Figure 13d,e. The laptop connected to the radar was a Dell G3 model with NVIDIA GTX 1650 4 GB, which has a 896 CUDA core.

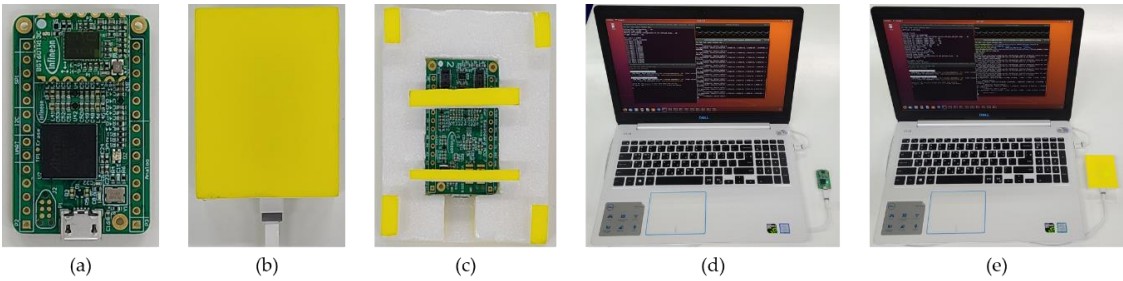

|     |     |     |     |     |
| --- | --- | --- | --- | --- |
| (a) | (b) | (c) | (d) | (e) |

**Figure 13.** Hardware for the hand gesture recognition system. (**a**) 60 GHz FMCW radar module; (**b**) radar module with case (front); (**c**) radar module with case (back); (**d**) radar–laptop connection without case; (**e**) radar-laptop connection with case.

Source and target domain data collection and experiments for gesture recognition were performed in various environments, as shown in Figure 14. Source domain data were collected only at the office, at which time the radar case was not installed. Each participant made 7 gestures at a height of about 10 cm from the radar. The source domain data are transformed into RDM through the signal processing in Section 2.1.2, and trained through deep learning algorithms in Section 3. Target domain data were collected in a lecture room, corridors, etc., including an office, and a case was mounted on the radar in order to make sure there were changes in data between domains. Likewise, participants not included in the source domain data collection made gestures at a height of about 10 cm from the radar. The target domain data are also transformed into RDM through signal processing, and the gesture result is inferred by being input to the deep learning network including the domain discriminator. Details of the collected data are described in Section 4.3, and the classification results accordingly are shown in Sections 5.1 and 5.2.

An experiment to confirm the real-time performance of the proposed system was conducted in an exhibition hall. In the above data collection process, all participants sat down to conduct the experiment, whereas in the real-time experiment, all participants stood and made a gesture. The results for the real-time experiment are detailed in Section 5.3.

The software used in the experiment and the interaction between the software are described in Section 4.2. Additionally, a demo video about the experiment setting for the data collection and real-time experiment can referred to in [51].

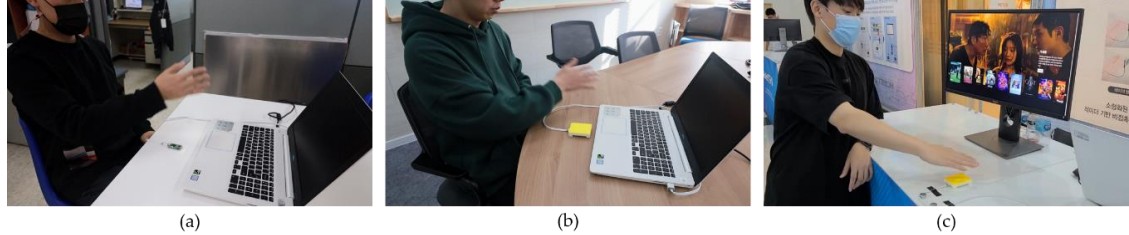

**Figure 14.** Examples of various experimental environments: (**a**) office for collecting source domain data; (**b**) lecture room for collecting target domain data; (**c**) exhibition hall for real-time experiment.

### 4.2. Implementation

In order to extract the reflected raw signal received by the antennas of the FMCW radar, C language-based software was used. This software includes Infineon-manufactured headers required for radar operation, and allows the laptop connected to the FMCW radar to extract the signal data stored in the radar's small buffer through the serial port. Through the FMCW radar setting, one chirp signal has a bandwidth of 6 GHz (from 57.5 to 63.5 GHz). One shape group contains 16 chirp signals, and the software extracts data in units of 4 shape groups. That is, 4 shape group data that are simultaneously received from three receiving antennas are extracted and a total of 192 reflected chirp signals are stored in the laptop.

C++ language-based software was used to process chirp signals stored in a laptop connected to the FMCW radar. This signal processing software includes the 2D FFT, CFAR, background subtraction, and RDM generation processes described in Section 2.1.2. NVIDIA's cuFFT library was used to accelerate computation through GPU processing of 2D FFT, and OpenCV's MOG2 algorithm was used for background subtraction. In addition, by implementing the processes from Equations (5) to (10), only the RDM judging that the gesture has occurred is stored.

The proposed AI model, as shown in Figure 8, for training and testing RDM input for gestures was implemented based on Python and PyTorch libraries. This Python-based AI model is connected to signal processing software through socket communication. For the 3D-CNN-based feature extractor described in Section 3.3, the basic block employing the Inception structure shown in Figure 12 was implemented using the Conv3D class of PyTorch. In addition, each class in torch.nn was used for batch normalization, dropout, and activation functions. The basic building blocks of torch.nn were used to implement the fully connected layer and cross-entropy loss function used in the gesture recognizer and domain discriminator described in Sections 3.4 and 3.5, respectively.

Parameters for the 3D-CNN-based learning network proposed in Section 3 were heuristically set that show optimal performance for the source domain through a number of experiments. In consideration of the gesture recognition performance and the time required to derive the classification result, the 2D FFT size was set to $512 \times 256$. The batch size, epochs, learning rate, dropout rate, and momentum for optimizing the network parameters were set to 10, 500, $10^{-5}$, 0.5, and 0.9, respectively. Finally, adaptive moment estimation (Adam) was used as the optimizer.

To evaluate the possibility of real-time utilization of the proposed system, an over the top (OTT) service prototype that can be interworked with the above three software was implemented. This OTT web application was implemented with the React web framework; the web page is manipulated by gesture recognition results using Selenium, an automated web testing framework in Python. Each gesture class is mapped to an interface for left/right, enter, play, and stop. Finally, users can choose videos and play with the real-time hand gesture interface.

### 4.3. Dataset

To evaluate the performance of the proposed hand gesture recognition system, the datasets of 7 types of hand gestures were collected: (1) swipe left (SL), (2) swipe right (SR), (3) clockwise rotation (CR), (4) counter-clockwise rotation (CCR), (5) grab (G), (6) push (P), and (7) double push (DP) as

shown in Figure 15. Signal samples were collected through two recording sessions, 30 times per gesture, from 10 participants in order to establish a source domain dataset. Ten participants of various ages consisted of 3 females and 7 males, and had different physical conditions (height, weight, posture using system, and distance to the radar, etc.) as well as gesture characteristics. The only constraint during sample collection was to make the gesture within a distance of about 10~15 cm from the radar so that the hand gesture could be recognized. Through this process, 4200 samples to construct the source domain dataset were collected.

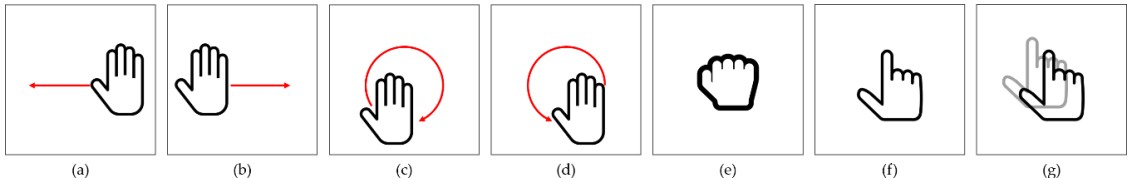

**Figure 15.** Seven hand gestures for the proposed system: (**a**) swipe left (SL); (**b**) swipe right (SR); (**c**) clockwise rotation (CR); (**d**) counter-clockwise rotation (CCR); (**e**) grab (G); (**f**) push (P); (**g**) double push (DP).

In addition, to construct the target domain dataset, the above sampling process was the same, but samples were collected from 10 new participants who did not participate in the source domain data collection process. In order to sample various types of data, unlike the source domain dataset that collected data without a case mounted, the case was not mounted in the first session of the target domain data collection process, and the case was mounted in the remaining sessions. The collected raw signal is attenuated or vanished when the reflected signal is very weak due to the case blocking the FMCW radar antennas. This new group of target domains consisted of 2 females and 8 males with different physical conditions at different ages. Through this process, 4200 samples were also collected to construct a target domain dataset.

### 4.4. Compared Gesture Recognition Algorithm

We compared the proposed system with existing radar-based gesture recognition systems using other machine learning algorithms to verify the classification accuracy performance. First, a motion profile-based recurrent neural network (RNN) encoder was used to measure the gesture recognition accuracy. The structure of the RNN encoder is simple, but it is suitable for learning the derived 2D motion profile. The key insight of using the RNN encoder is to utilize the temporal locality of the motion profile within a short period when a gesture is taken. For this, the motion profiles when the gesture is input are extracted using Equations (15) to (17). The extracted motion profiles are input to an RNN encoder with one hidden layer and 128 nodes. Since motion profiles for the $K$ periods in which the gesture occurs are extracted by the CFAR algorithm, the entire ENN structure has $K$ stages. The result of the input gesture is inferred through the softmax layer after the RNN in the last step. In the training stage of the RNN encoder, the batch size, epochs, learning rate, epochs, dropout rate, and momentum were set to 10, 500, $10^{-5}$, 0.5, and 0.9, respectively. This is the same as the values of the hyperparameters used in the proposed system, and Adam was also used as the optimizer.

In addition to the RNN encoder-based machine learning algorithm, a gesture recognition system based on 2D-CNN [17] was also compared. This system used a Doppler spectrogram expressed in the time–frequency format as input data. This is compared to the proposed system using RDM, which is expressed in terms of radial velocity and distance. Through this, the effect of the type of input data on the classification result can be analyzed. In addition, it is possible to grasp the accuracy of inference according to the difference between 2D-CNN and 3D-CNN structures. This 2D-CNN-based system used down-sized Doppler spectrograms and extracted features using a $5 \times 5$ convolution filter. Similar to the proposed system, the overall CNN structure includes three convolution layers. However, it is mentioned that 5–20 convolution filters were used for each layer. For performance comparison,

the number of convolution filters per layer was adjusted to approach the 93.1% accuracy derived by the authors of the paper. Each convolution layer is combined with an activation function and a pooling layer. ReLU is used as the activation function, and the results are finally classified through one fully connected layer. Like the RNN encoder, the hyperparameters of the 2D-CNN are set the same as those of the proposed system. However, the optimizer used stochastic gradient descent (SGD), as mentioned in the 2D-CNN-based system.

## 5. Performance Evaluation

### 5.1. Offline Test with Source Domain Dataset

The accuracy of hand gesture recognition of the proposed system is measured through five-fold cross-validation using the source domain dataset to confirm that the feature extractor and gesture recognizer operate properly. For this, the source domain dataset was divided into five groups, and four sets were used as training data and one set as testing data. Table 1 shows the gesture recognition accuracy of three different systems derived through 10 learning processes for each fold. The proposed system, taking into account the spatial–temporal features according to gestures, showed the highest recognition accuracy at about 99%. Both RNN and 2D-CNN-based gesture recognition accuracy showed more than 90% stable performance. Through this, it can be confirmed that each machine learning algorithm is properly implemented and classifies data in the source domain accurately.

**Table 1.** Accuracy (%) comparison per fold and the average of the proposed hand gesture recognition system with other machine learning-based systems using the source domain dataset.

|  | **Fold-1** | **Fold-2** | **Fold-3** | **Fold-4** | **Fold-5** | **Avg. (std.)** |
|---|---|---|---|---|---|---|
| RNN | 91.72 | 90.19 | 89.98 | 90.52 | 88.97 | 90.27 (±0.89) |
| 2D-CNN | 92.54 | 94.41 | 93.53 | 91.83 | 95.61 | 93.58 (±1.34) |
| 3D-CNN | 98.79 | 99.14 | 99.07 | 99.38 | 98.91 | 99.06 (±0.20) |

Figure 16 illustrates the accuracy of gesture recognition for each participant. All recognition systems show an average accuracy of over 90%, but the proposed system shows high accuracy and low deviation. Figure 17 shows the gesture recognition confusion matrix for each machine learning algorithm, which means that RNN and 2D-CNN cannot properly classify directional motions such as swipe and rotation, or motions in which spatial information changes over time such as push and double push. On the other hand, the proposed system generally shows high classification performance for all hand gestures.

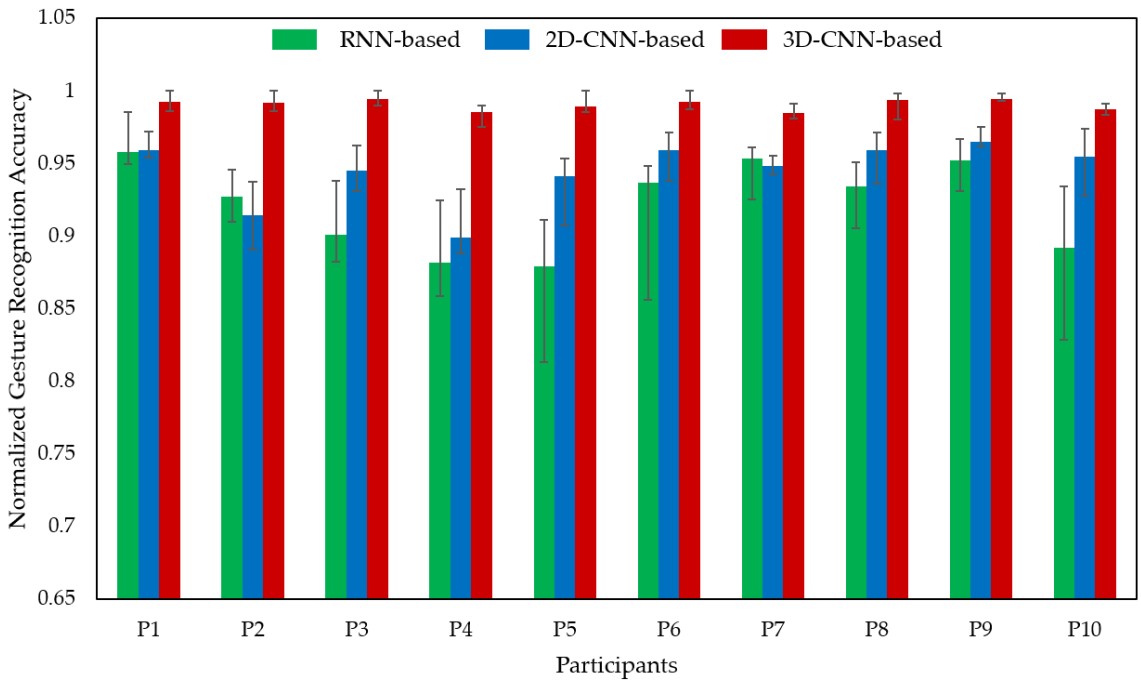

**Figure 16.** The gesture recognition accuracy of three system for each participant.

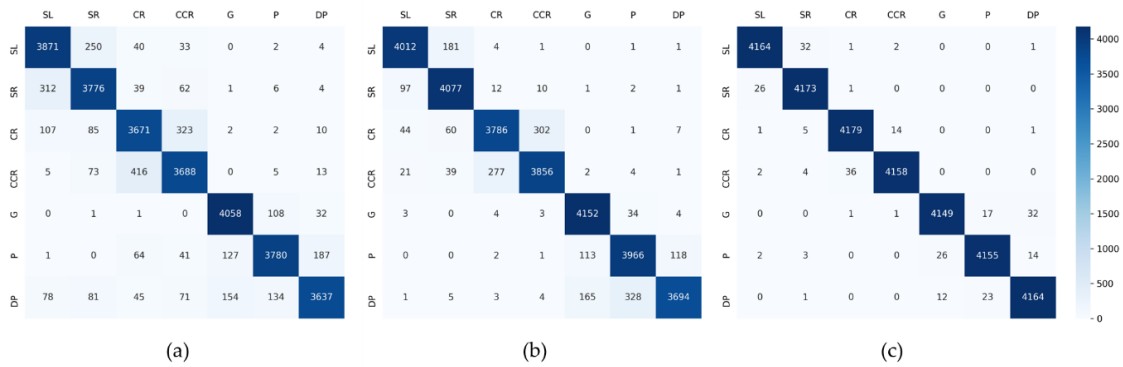

**Figure 17.** The confusion matrices of each machine learning for the source domain dataset. (**a**) Recurrent neural network (RNN); (**b**) 2D-CNN; (**c**) 3D-CNN.

*5.2. Offline Test with Target Domain Dataset*

Similar to the offline test using the source domain dataset, five-fold cross-validation using the target domain dataset was performed. This evaluation is to confirm whether the proposed system can properly classify target domain data with different features by a new participant. In the target domain dataset, the features collected are changed due to the new participant's gesture, the presence of a radar case, and the data sampling location. Unlike the previous test process, the performance of the proposed system without the domain discriminator was also measured to verify the effect of domain adaptation. Table 2 represents the classification accuracy of the four gesture recognition systems.

**Table 2.** Accuracy (%) comparison per fold and the average of the proposed hand gesture recognition system with other machine learning-based systems using the target domain dataset.

|  | Fold-1 | Fold-2 | Fold-3 | Fold-4 | Fold-5 | Avg. (std.) |
|---|---|---|---|---|---|---|
| RNN | 81.61 | 77.63 | 78.16 | 76.82 | 80.85 | 79.01 (±1.87) |
| 2D-CNN | 72.12 | 73.59 | 75.40 | 71.48 | 71.65 | 72.85 (±1.47) |
| 3D-CNN | 98.66 | 98.75 | 99.10 | 98.55 | 99.04 | 98.82 (±0.21) |
| (w/o DA) | 84.47 | 82.18 | 83.67 | 81.80 | 83.58 | 83.14 (±0.99) |

The proposed system to which domain adaptation is applied showed a very high recognition accuracy of 98.82% for the target domain dataset. RNN and 2D-CNN-based machine learning algorithms have significantly reduced the classification accuracy for the target domain dataset. Three-dimensional CNN to which domain adaptation is not applied could prevent sharp performance degradation due to proper classification of spatial–temporal features according to hand gestures. In the proposed system, the domain discriminator works properly in an adversarial manner, and it can be seen that gestures are accurately recognized by extracting domain-independent features for target domain data.

Figure 18 illustrates the accuracy of gesture recognition for each participant in the target domain group. The sharp degradation of 2D-CNN is noticeable, and the biggest factor for this is that similar Doppler spectrograms are extracted from different motions depending on the participant. On the other hand, RNN and 3D-CNN without the domain discriminator showed an accuracy of about 80% because the distinction was relatively clear. In addition, the proposed system showed the highest recognition accuracy because it can extract features only related to gestures, that is, it cannot distinguish from which domain the features extracted from the classifier are generated.

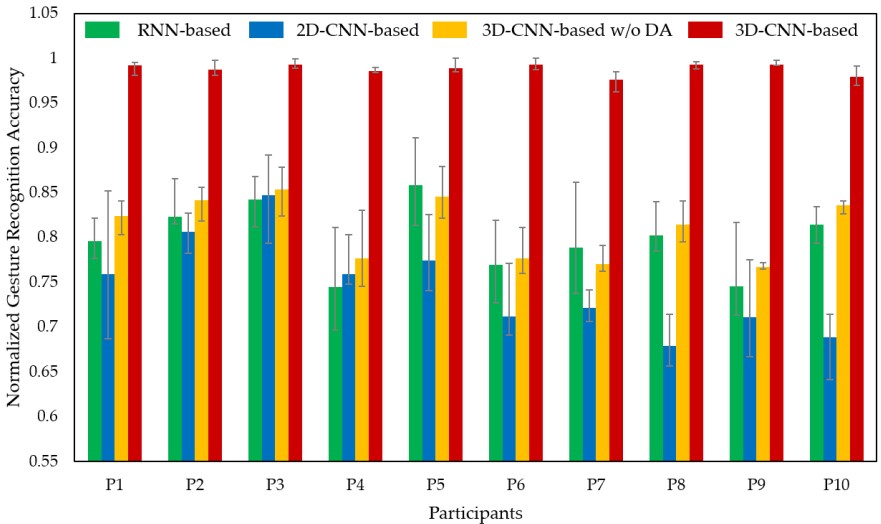

**Figure 18.** The gesture recognition accuracy of three system for new participants (target domain).

The confusion matrices of four machine learning algorithms are shown in Figure 19. In the case of using the target domain dataset, the feature map is changed not only by the gesture but also by the surrounding environment or user. Then, the result of misclassification is represented regardless of the gesture type. Despite these difficulties, the proposed system showed very little misclassification only between similar operations (e.g., CW–CCW, SL–SR, and P–DP), and overall high classification accuracy.

## 5.3. Online Test

We implemented an application that can control an OTT streaming service using the proposed hand gesture recognition system and tested it in a real environment, as shown in Figure 20. For this, the hyperparameters were adjusted and the domain discriminator was optimized so that the response speed to recognize gestures was within 0.5 s so as not to degrade the user experience. The OTT control service based on the proposed system was exhibited at the demonstration to confirm the performance of gesture recognition in various environments, for example, a crowded situation, a case of several people existing near the radar, etc. In the demo process, 50 different users participated, and each user entered 17–22 gestures into the OTT application. Through this, a total of 992 input data were obtained. At this time, gestures taken by each user were separately recorded as ground truth in order to compare the results inferred by the proposed system. As a result, 901 gestures inferred by the proposed system matched the gestures actually taken by the user. Considering that some parameters are lighter to

guarantee real-time performance and target domain data are used, it can be seen that the inference success rate of 90.8% is quite high performance.

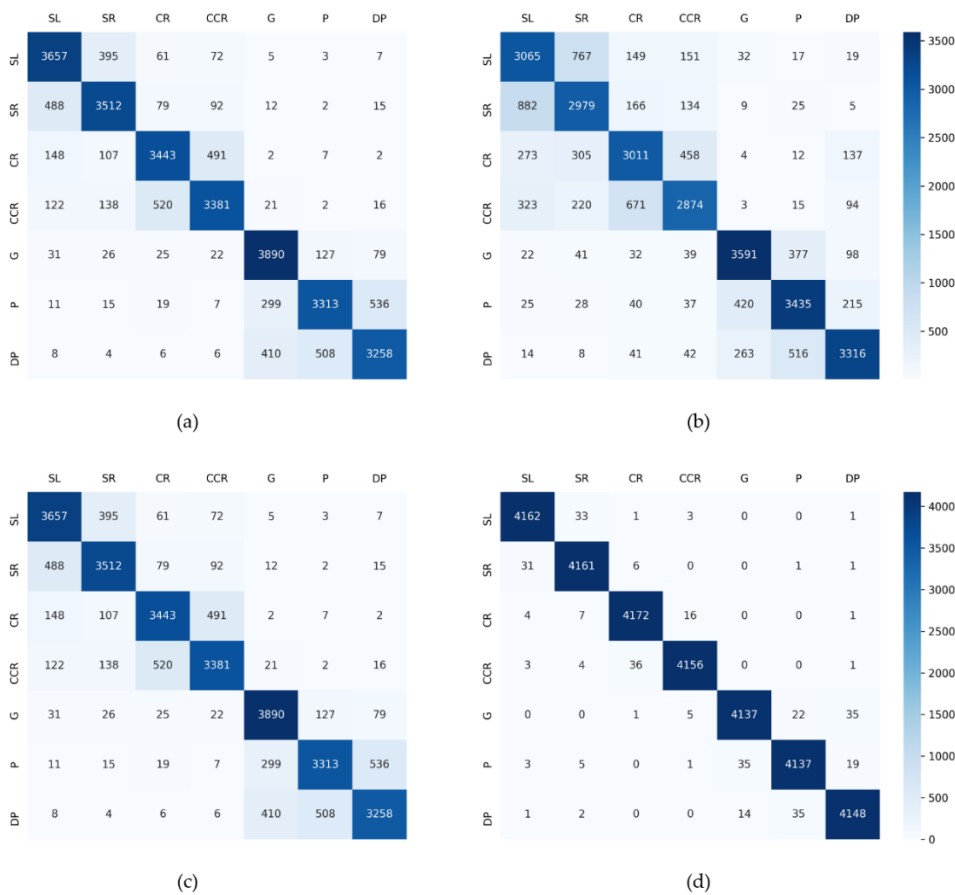

**Figure 19.** The confusion matrices of each machine learning for the source domain dataset. (**a**) RNN; (**b**) 2D-CNN; (**c**) 3D-CNN without domain discriminator; (**d**) 3D-CNN.

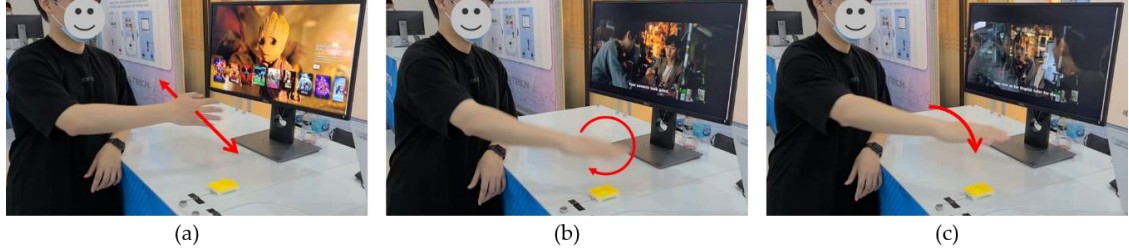

**Figure 20.** The over the top (OTT) control application based on the proposed gesture recognition system. (**a**) Swipe: scroll and forward/rewind; (**b**) clockwise rotation: go back; (**c**) push: play/stop.

## 6. Conclusions

This paper proposed a 60 GHz FMCW radar-based hand gesture recognition system and a method to improve recognition accuracy. Instead of the motion profile data adopted by Google Soli, the range–Doppler matrix was used as an input to extract more precise spatial–temporal features for hand gestures. However, this type of input requires a large amount of computation and is not efficient for the gesture recognition system. The proposed system introduced a 3D-CNN with an Inception structure to process the sequence of the range–Doppler matrix. Thus, it accelerated the feature extraction process. Through the improved 3D-CNN-based machine learning network, precise spatial–temporal features of various hand gestures could be extracted.

In addition, motion recognition systems have a common problem, which is that recognition accuracy is degraded due to differences in features between training data and actual test data. In other words, the difference between the user and the surrounding environment in the training data collection stage and the actual use stage occurs. It causes different feature extraction for the same gesture. Therefore, the classifier cannot properly infer the target gesture. To solve this problem, in this paper, a domain adaptation algorithm is applied to minimize the discrepancy between the source domain and the target domain. The domain discriminator was implemented to learn the loss of domain classification in an adversarial manner so that the two domains were not distinguished as much as possible. Finally, domain-independent features were extracted from the input signal so that the learning network using source domain data could operate properly. Then, improved recognition accuracy could be achieved.

Gesture samples were collected from two groups with 20 users (one in the source domain and the other in the target domain) to verify the classification accuracy of the proposed hand gesture system. As a result, the classification accuracy of the proposed system was 98.8% on average for the target domain dataset, while the accuracies of other machine learning algorithms were lowered. In the real-time online test using an optimized network to improve the response speed of the proposed system, the hand gestures of 50 users were successfully recognized with an accuracy of about 90%.

The biggest advantage of the proposed system is that it can maximize the generalization performance with only a small number of training data. Furthermore, it is also an advantage to be able to extract elaborate spatiotemporal features within a relatively fast time compared to the existing 3D-CNN. Features extracted through the proposed convolution process are more effective in inference than the motion profile or Doppler spectrogram. These advantages were demonstrated through comparative experiments and service application.

In future work, we intend to design a lightweight learning network while ensuring decent classification accuracy. After that, the unsupervised-based domain adaptation most suitable for this learning model is designed and applied. Accordingly, it will ensure maximum real-time classification accuracy without adjustment of hyperparameters. Furthermore, the domain adaptation module will function sufficiently to further reduce the gap between the source domain and the target domain.

**Author Contributions:** Conceptualization and methodology, H.R.L.; software, H.R.L.; validation, H.R.L. and J.P.; formal analysis, H.R.L.; investigation, H.R.L. and J.P.; resources, H.R.L.; data curation, H.R.L.; writing, review and editing, H.R.L. and J.P.; supervision, Y.-J.S. All authors have read and agreed to the published version of the manuscript.

**Funding:** This work was partly supported by Institute for Information and Communications Technology (IITP) grant funded by the Korean government (MSIP) (No. 2019-0-01906, Artificial Intelligence Graduate School Program (POSTECH)) and Regional Demand-Specific R&D Support Program from Ministry of Science and ICT (Republic of Korea) (CN19100GB001).

**Conflicts of Interest:** The authors declare no conflict of interest.

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
