# Peer review of "Improving Classification Accuracy of Hand Gesture Recognition Based on 60 GHz FMCW Radar with Deep Learning Domain Adaptation"

_electronics, doi:10.3390/electronics9122140_

Round 1

Reviewer 1 Report

This is a very interesting problem, but certain very important details are not presented.

(1) What is the actual bandwidth of the chirp signal used for experiments?

(2) Detection algorithm and AI models are described as general concepts. Most of the technical details are not presented in full detail. The presented material is not enough to duplicate the same experiment.

(3) Which SW is used to acquire the raw data?

(4) Which SW is used to train the AI model?

(5) Is the final AI based recognition system runs in real-time? Which SW is used for that purpose? 

In general, the presented material is not enough to duplicate the same experiment. I don't see any MATLAB, Python or any AI library name (Tensorflow, PyTorch). 

Author Response

Thank you for your valuable review comments. It was very helpful in elaborating the paper.

Point 1: What is the actual bandwidth of the chirp signal used for experiments?

Response 1: The actual bandwidth of the chirp signal is 6 GHz (from 57.5 to 63.5 GHz) for experiments. The 6 GHz is the bandwidth of the maximum chirp signal supported by the FMCW radar used in the experiments. This is described in the newly added Section 4.2 Implementation part.

Point 2: Detection algorithm and AI models are described as general concepts. Most of the technical details are not presented in full detail. The presented material is not enough to duplicate the same experiment.

Response 2: In order to reproduce the experiment using the proposed system, details on the implementation have been added to Section 4.2. A total of 3 software is required for the experiment, and this includes raw signal extraction, signal processing and AI model. Section 4.2 contains the contents of the role of each software and the libraries used, and it is described in detail in the following comment points.

Point 3: Which SW is used to acquire the raw data?

Response 3: For raw signal data extraction, C language-based program was implemented and used. The FMCW radar stores the reflected and returned raw signal in a small buffer, and the laptop connected to the radar extracts it through the serial port. 192 chirp signals received from three antennas are stored in the laptop every frame by using the implemented signal extraction software. In addition, the raw signal data stored in the laptop is processed into RDM through signal processing for input into the AI model, which is done through software written in C++ language. This signal processing software uses NVIDIA’s cuFFT library for 2D FFT acceleration, and OpenCV’s MOG2 algorithm for background subtraction. Furthermore, this software contains the equations described in Section 2.1.2 for RDM conversion. This process is described in Section 4.2.

Point 4: Which SW is used to train the AI model?

Response 4: Python was used to implement the 3D-CNN-based AI model described in Section 3.1 to train and classify the extracted RDM input. In particular, PyTorch was actively utilized to implement the proposed basic block for enhanced 3D convolution and to utilize various layer such as fully connected layer, dropout, activation function, etc. In more detail, the basic block for improved 3D-CNN has been newly defined using the existing Conv3D class of torch.nn, and various layers such as ReLU, BatchNorm, and MaxPool connected to this basic block were implemented through basic building class (block) of torch.nn. This AI model is connected through socket communication with the signal processing software, and finally trains the extracted RDMs or infers the results. This process is also described in Section 4.2.

Point 5: Is the final AI based recognition system runs in real-time? Which SW is used for that purpose?

Response 5: The proposed hand gesture recognition system can operate in real-time. Since the AI model has a model trained with the source domain, and the domain adaptation process for the input hand gesture is performed without additional data training. Therefore, the proposed can show a fast gesture inference response. To confirm this, a prototype of an OTT control system similar to Netflix was implemented by using React web framework and Selenium, which is an automated web testing framework. This service application is linked with the AI model, and each hand gesture is mapped to a function required for media control such as left/right, select, play, and stop. This real-time application is described in Section 4.2, and as shown in the result of real-time gesture recognition (Section 5.3), it took an average 0.5 seconds from gesture input to media control completion.

Point 6: In general, the presented material is not enough to duplicate the same experiment. I don’t see any MATLAB, Python or any AI library name (Tensorflow, PyTorch)

Response 6: All of the review comments pointed out above appear to be applicable this point. Accordingly, Section 4.2 was newly added to describe how the system was implemented so that the experiments conducted in this paper can be reproduced. This implementation section contains the header information and essential libraries used to implement each software. Besides, the role of each software is explained step by step, and the relationship between the software is described. In addition, it contains the PyTorch class used to implement the AI model and hyper-parameters applied to the experiment. It is believed that the experimental process of this paper can be reproduced by using the layer arrangement and parameter described in Section 3 along with the contents of Section 4.2.

Reviewer 2 Report

This paper proposed a solution in the field of radar-based hand gesture recognition which tries to overcome the problem of recognition accuracy degradation resulting from even slight differences in movement for individual user. The authors applied the domain adaptation to minimize the differences among gesture information. To verify its effectiveness, domain discriminator that cheats the classifier was applied to a deep learning network.

Although the training sample in the experiment was rather small and consisted of only 10 participant who are source of data on 7 different hand gestures, the recognition accuracy of an average of 98.8% was a very good result while testing on hand gestures of 10 users that were not included in the training data. The domain adaptation algorithm proposed for the hand gesture recognition system proved successful, at least in the experiment.

The paper is sound technically and the methodology applied is acceptable. The first Section covers introductory literature review in the fields of vision camera based systems, motion based systems, the use of wireless communication, and FMCW radar modules and domain adaptation, however it is very superficial in terms of describing quoted sources and completely fails to cover the field of deep learning. This should be taken care of by elaborating on the sources the paper refers to in line 57 [references 16-21] and line 82 [references 22-31]. Furthermore, an introduction to the role and wide range of successful deep learning applications learning should be added in Introduction, including in particular recent papers (10.3390/electronics9111863, 10.3390/electronics9101712, 10.3390/electronics9020266, 10.3390/electronics9010135 and others). What is more, some sources included in the paper’s References seem oldish and it is advised that authors consider referring to fresher journal papers.

Preliminaries to radar-based gesture recognition and domain adaptation are covered in a good way, as well as the presentation of the experiment. However, it would be recommended that the authors elaborate more in Section 4.3 on other systems and algorithms their solution is compared to. Furthermore, the hyper-parameters used should be provided and discussed for better comprehensibility and reproducibility.

The results the paper refers to in lines 508-509 are not described well enough and need elaboration.

Conclusions do not include all most necessary points. That needs to be corrected, too, and I would also like to see clear statements on the advantages as well as future applications and study directions the author want to follow as a result of their results.

The paper needs language editing, in particular plural forms often omitted (e.g. application in line 13 or algorithm in line 420 etc.). Some sentences need to be rephrased to be shorter and thus clearer to the reader.

Author Response

Thank you for your valuable review comments. It was very helpful in elaborating the paper.

Point 1: This paper is sound technically and methodology applied is acceptable. The first section covers introductory literature review in the fields of vision camera based systems, motion based system, the use of wireless communication, and FMCW radar modules and domain adaptation, however it is very superficial in terms of describing quoted sources and completely fails to cover the field of deep learning. This should be taken care of by elaborating on the sources the paper refers to in line 57 [references 16-21] and line 82 [references 22-31]. Furthermore, an introduction to the role and wide range of successful deep learning applications learning should be added in Introduction, including in particular recent papers (10.3390/electronics9111863, 10.3390/electronics9101712, 10.3390/electronics9020266, 10.3390/electronics9010135 and others). What is more, some sources included in the paper’s References seem oldish and it is advised that authors consider referring to fresher journal papers.

Response 1: Introduction has been revised so that each superficially described reference can be briefly explained (line 75-83 and line 103-117). In the process, some oldish references have been deleted or replaced with fresher papers. In addition, a paragraph (line 52-66) explaining the role and range of deep learning was written by adding several references along with the papers recommended. This paragraph explains how deep learning can be applied to various applications to improve the performance of existing systems or solve problems more efficiently. Furthermore, it is described that deep learning is being actively used in the field of gesture recognition.

Point 2: It would be recommended that the authors elaborate more in Section 4.3 on other systems and algorithms their solution is compared to. Furthermore, the hyper-parameters used should be provided and discussed for better comprehensibility and reproducibility.

Response 2: To verify and compare the performance of the proposed system, we implemented two gesture recognition systems. One is based on the RNN encoder, and the other is based on the 2D-CNN structure. The RNN encoder, which has a relatively simple structure, can infer the result through the temporal locality of motion profiles according to gesture. This can be compared with the proposed system that infers results through spatiotemporal features. The RNN encoder has one hidden layer and one layer has 128 nodes. Motion profiles are generated during the period of gesture occurrence by the CFAR algorithm. Accordingly, the RNN encoder has the same number of stages as this period, and finally classifies the result by softmax layer. The 2D-CNN based system is similar in structure to the proposed system. However, Doppler spectrograms are used as input data and 2D convolution filter is used. Through this, it is possible to analyse the effect of the type of input data and the dimension of the convolution filter on the classification performance. The three convolution layers are combined with the ReLU activation function and pooling layer along with several convolution filters. Finally, the result is inferred through one fully connected layer. The network structure of each system and the setting of hyper-parameters have been described by supplementing in Section 4.4. It seems that it is possible to reproduce the two systems used in the comparative experiments using the contents.

Point 3: The results the paper refers to in line 508-509 are not described well enough and need elaboration.

Response 3: A lightweight AI model and OTT application were used to verify the real-time performance of gesture inference of the proposed system. In this experiment, 50 different users participated, and about 20 gesture data were collected for each user. A total of 992 gesture data were collected during experiment, and the proposed system successfully inferred 901 gestures. As a result, real-time classification performance of 90.8% was achieved. The elaborated real-time experiment is described in Section 5.3.

Point 4: Conclusion do not include all most necessary points. That needs to be corrected, too, and I would also like to see clear statements on the advantages as well as future applications and study directions the author want to follow as a result of their results.

Response 4: The biggest advantage of the proposed gesture recognition system is that it can maintain high generalization performance by using domain adaptation. This is possible because only spatiotemporal features that are independent of the user or environment are extracted. It was proved by showing high classification performance for different gesture data of new users. The relatively fast 3D convolution filtering is also an advantage. It has been proven by showing that it can be used in real-time with the adjustment of a few hyper-parameters. Nevertheless, 3D-CNN requires a relatively large amount of computation, so designing a more lightweight AI model is planned as a future work. Researching suitable unsupervised domain adaption is also considered as a future work. In the Conclusion section, statements of advantages and future plans were added.

Point 5: The paper needs language editing, in particular plural forms often omitted (e.g., application in line 13 and algorithm in line 420 etc.) Some sentences need to be rephrased to be shorter and thus clearer to the reader.

Response 5: Including the examples pointed out, errors about the plural form were corrected. In addition, some ambiguous sentences have been explained to make them easier to understand. Very long sentences were also rephrased into short sentences. Unnecessary repeated phrases and expressions have been removed.

Round 2

Reviewer 1 Report

Lots of implementation details are missing. If someone would like to replicate these for comparison with a similar setup/algorithm, it will be very difficult. Most details are still very sketchy.

I strongly encourage authors to record demo experiments (and their test setup) and share it on YouTube or similar platforms.

Author Response

Point 1:  Lots of implementation details are missing. If someone would like to replicate these for comparison with a similar setup/algorithm, it will be very difficult. Most details are still very sketchy. I strongly encourage authors to record demo experiments (and their test setup) and share it on YouTube or similar platforms.

Response 1: According to the comments, we have added information about the experimental settings to Section 4.1. In addition, a video about settings and a real-time experiment that can support this explanation has been uploaded to YouTube. A YouTube URL to this has been added as reference [51].